# Business Environment, Attitudes and Entrepreneurial Intentions as Antecedents of Entrepreneurial Inclination among University Students

**Maira Rafaela Vargas-Martínez** [1,2,*]**, Joselina Caridad Tavarez-De Henríquez** [2]**, Nirda de Jesús Colón-Flores** [2] **and Cándida María Domínguez-Valerio** [2]

[1] Department of Statistics, Econometrics, Operations Research, Business Organization and Applied Economics, Universidad de Córdoba, 14071 Córdoba, Spain

[2] Department of Economic and Social Sciences, Universidad Tecnológica de Santiago, Santiago 5100, Dominican Republic; jtavares@utesa.edu (J.C.T.-D.H.); nirdacolon@docente.utesa.edu (N.d.J.C.-F.); candidadominguez1@docente.utesa.edu (C.M.D.-V.)

\* Correspondence: mairav@utesa.edu

**Abstract:** Entrepreneurship education has become increasingly relevant. For some years now, the business environment for starting a new company in the Dominican Republic has been considered very propitious. This has caused many universities to incorporate training in entrepreneurship into their study plans. This study aims to analyse whether the cognitive and affective components, the country's business environment, university training in entrepreneurship and attitudes towards entrepreneurship are antecedents of the entrepreneurial intention of university students. It also seeks to investigate the relationship between entrepreneurial intention and entrepreneurial behaviour of students. Data were collected through a structured questionnaire from a sample of 523 students who had taken courses on entrepreneurship. The data have been analysed through the SPSS and Smart-PLS programs. The results of this research highlight the importance of the affective component and attitudes towards entrepreneurship in the formation of students' entrepreneurial intentions. This reinforces the importance of cultivating positive attitudes through educational interventions. It is also worth noting the influence of students' entrepreneurial intentions on entrepreneurial inclinations. Finally, it is worth highlighting the high predictive power of the entrepreneurial intentions variable and more specifically the attitudes towards entrepreneurship variable as responsible for 35.38% of the variability of entrepreneurial intentions. These results contribute to the understanding of the factors that drive entrepreneurial intentions among university students and provide a foundation for future research.

**Keywords:** business environment; attitudes; entrepreneurship; behaviors; university

## 1. Introduction

Entrepreneurship has played a significant role in the economic prosperity and social stability of many developed countries [1]. In this regard and for some years now, the business environment for starting a new firm in the Dominican Republic has been considered very conducive [2,3]. This has led many of the country's universities to promote entrepreneurship training as a cross-cutting educational offer in any area of study. For example, some universities have included compulsory entrepreneurship subjects in all curricula, regardless of the degree programme. In other words, a student of business administration will be trained in entrepreneurship, but so will a student of law, engineering or medicine. However, it is crucial to determine the success of these initiatives towards entrepreneurship education at a university, as university entrepreneurship education has been found to improve students' attitudes towards entrepreneurship [4].

Entrepreneurship is a multifaceted phenomenon [1] that is defined as an individual establishing and managing a business for profit and growth [5]. Thus, the cognitive component, which consists of students' beliefs, thoughts and knowledge about an attitude construct, plays an essential role in entrepreneurship [1]; likewise, the affective component, which refers to the feelings and moods people experience, influences various aspects of entrepreneurial inclinations [6]. Similarly, both a country's business environment and entrepreneurship education programmes can influence the intention to start a business [7]. In the case of the entrepreneurial environment, factors such as weak institutional environments, supportive infrastructure and favourable entrepreneurial climates may have an impact on students' entrepreneurial intentions [8]. In turn, university training in entrepreneurship promotes proactivity, innovation and creativity, which in turn influence entrepreneurial intention [9]. In addition, attitudes towards entrepreneurship are also considered essential predictors of entrepreneurial intention [10]. It has even been suggested that positive and negative perceptions of entrepreneurship are more important than cognitive factors in influencing entrepreneurial intention [11]. Similarly, it has been suggested that entrepreneurial intention influences students' entrepreneurial inclinations [12]. In this context, university engagement plays a fundamental role for the development of entrepreneurial skills, either from university curricula or through training programmes in companies [13].

By understanding the interaction between cognitive and affective components, a country's entrepreneurial environment, university education in entrepreneurship or entrepreneurial attitudes can provide valuable insights into the factors that drive entrepreneurial intentions and inclinations. Therefore, this research aims to find out whether the cognitive, affective, entrepreneurial environment, university training in entrepreneurship and entrepreneurial attitudes components are antecedents of the entrepreneurial intention of university students; in turn, it also seeks to find out the relationship between entrepreneurial intention and entrepreneurial inclinations of university students. This analysis also has the purpose of finding out the predictive power and the explained variance of the relationships established between the variables of this research. The findings of the study may help university managers to promote specific aspects of entrepreneurship education in the Dominican Republic, as educational policy makers need a deep understanding of the aspects that contribute to higher entrepreneurial intentions; this research can also cooperate to identify the backgrounds that students are most interested in when starting new businesses, especially because previous studies have presented some limitations due to the small sample size of students and the small number of variables used to predict the phenomenon of entrepreneurial intentions [14,15].

## 2. Theoretical Framework and Hypotheses

This section is split into three parts. The first part presents the relationship between the cognitive and affective components of entrepreneurial intention. The second part shows the relationship between the entrepreneurial environment, university training in entrepreneurship and attitudes towards entrepreneurship and entrepreneurial intention. Finally, the last part presents the relationship between entrepreneurial intention and entrepreneurial inclinations.

### 2.1. Cognitive and Affective Component of Entrepreneurial Intention

Cognitive psychology explores how mental processes evolve and change as individuals interact with others and their environment [16]. In this regard, the mental processes that occur within individuals have a relationship with the process of entrepreneurship [16]. Thus, cognitive processes play a crucial role in the formation of entrepreneurial intentions, since it has been argued that opportunities arise from individuals' intentions, which are derived from their cognitive processes [16]. Furthermore, affectation has been shown to influence various aspects of cognition and behaviour [17]. In the context of entrepreneurship, it has been suggested that affectation influences elements of the entrepreneurial process,

such as opportunity recognition and resource acquisition [17]. In the context of social entrepreneurship, cognitive modelling has been found to influence intention to engage in social entrepreneurship [18]. Based on the above, the following hypothesis is proposed:

**Hypothesis 1 (H1).** *The cognitive component (COG) influences entrepreneurial intention (EI).*

Affectation influences various aspects of cognition and behaviour [17]. Thus, positive affectation influences the generation of entrepreneurial ideas and intentions to pursue these ideas [19]. In addition, it has been suggested that the ability to understand the views of others and to react emotionally to the suffering of others stimulates the intention to help through entrepreneurial initiatives [20]. Furthermore, positive affective traits have a positive impact on attitude towards entrepreneurship, perceived behavioural control of entrepreneurship and social norms towards entrepreneurship, whereas negative affective traits have a negative influence on attitude and social norms [21]. In this context, people evaluate the same feelings and emotions differently due to their motivation, personality, past experience, reference group and unique physical conditions [22], which means that some people (students) may have positive feelings towards entrepreneurship education, while others might respond with an adverse reaction [1]. In the context of social entrepreneurship, the affective model has been found to influence the intention to engage in social entrepreneurship [18]. Based on the above, the following hypothesis is proposed:

**Hypothesis 2 (H2).** *The affective component (AFE) influences entrepreneurial intention (EI).*

*2.2. Business Environment, University Training in Entrepreneurship and Attitudes towards Entrepreneurship on Entrepreneurship Intention*

Aidis et al. [23] analysed entrepreneurial development in Russia in comparison with Brazil and Poland and found that the business environment and networks contribute to the relative advantage of internal entrepreneurs over external entrepreneurs in terms of new business creation. This suggests that a country's institutional environment can have a significant impact on entrepreneurial intention. Schawarz et al. [24] found that general attitudes, entrepreneurship, perceptions of the university environment and regional infrastructure significantly influenced students' interest in entrepreneurship. Contextual factors, such as business policies, programmes and infrastructure, have also been identified as important in creating a favourable business climate in a country [8]. Furthermore, the entrepreneurial environment has a direct impact on the ease of starting and managing entrepreneurial projects, which in turn influences entrepreneurial intentions [25]. Internal elements of the business environment, such as management practices, also play a role in shaping entrepreneurial intentions [25]. Therefore, and based on the above, the following hypothesis is put forward:

**Hypothesis 3 (H3).** *The country's business environment (ERD) influences entrepreneurial intention (EI).*

Wu and Wu [26] analysed the relationship between higher education and entrepreneurial intentions of university students, concluding that the diversity of educational backgrounds offered plausible explanations for the differences in entrepreneurial intentions among Chinese university students. Saeed et al. [27] highlighted the importance of education, training and business support in developing entrepreneurial skills. Karimi et al. [28] indicated that both elective and compulsory entrepreneurship education programmes had significant positive impacts on students' entrepreneurial intentions. According to Trivedi [29], the university environment can either greatly motivate students' entrepreneurial spirit or create obstacles for them. This author emphasised that if universities do not provide the necessary knowledge, resources and support services for start-ups, students' entrepreneurial intentions may diminish. Koe [9] analysed the influence of entrepreneurial orientation on entrepreneurial intentions among university students, finding that students who demon-

strated entrepreneurial intention were positively influenced by their proactivity, creativity and innovativeness. Therefore, entrepreneurial education can positively influence entrepreneurial intentions [30]. Based on the above, the following hypothesis is put forward:

**Hypothesis 4 (H4).** *University training in entrepreneurship (PRO) influences entrepreneurial intention (EI).*

Gender, age, parental entrepreneurial background, prior entrepreneurial and work experience and participation in entrepreneurship programmes can affect students' attitudes towards entrepreneurship and their intention to become entrepreneurs [31,32]. For example, students with family and personal experience in entrepreneurship tend to have a more positive attitude towards entrepreneurship as a career option [33]. In addition, factors such as social stability can affect entrepreneurial intention [34]. Overall, studies have shown that a positive entrepreneurial attitude and willingness to start a business influence future business intention [35]. In this regard, the level of entrepreneurial education students receive is positively related to their entrepreneurial intention [36]. Higher education can therefore play a relevant role in shaping attitudes towards entrepreneurship, particularly through training and experiences that enhance the viability of entrepreneurship in students [37]. In their study, Souitaris et al. [38] showed that the programs increase some attitudes and general entrepreneurial intention and that inspiration is the most influential benefit of the programs. It has even been suggested that entrepreneurship education may mediate the relationship between entrepreneurial attitudes and entrepreneurial intentions [39]. Based on the above, the following hypothesis is put forward:

**Hypothesis 5 (H5).** *Attitudes towards entrepreneurship (ATE) influence entrepreneurial intention (EI).*

*2.3. Entrepreneurial Intention and Entrepreneurial Inclinations*

The theory of planned behaviour defines entrepreneurial intention as people's willingness to engage in entrepreneurial inclinations [40]. This theory explains that there is a positive effect of entrepreneurial intention on entrepreneurial inclination, which has been confirmed in different studies [41–43]. Yang [41] tested the validity of this theory to predict entrepreneurial intention among university students. The study suggested that attitude was the most effective predictor of entrepreneurial intention, followed by subjective norms and perceived behavioural control. Gender and parental entrepreneurial experience also had a significant impact on entrepreneurial attitude, subjective norms, perceived behavioural control and entrepreneurial intention. Yang [41] also highlighted the role of effective entrepreneurship education in improving perceived behavioural control and entrepreneurial intention. In this context, entrepreneurial intention has been widely associated with an individual's willingness to develop entrepreneurial inclinations and engage in starting a new business [12,13,44–47]. Based on the above, the following hypothesis is put forward:

**Hypothesis 6 (H6).** *Entrepreneurial intention (EI) influences entrepreneurial inclinations (COM).*

The proposed structural model is presented in Figure 1.

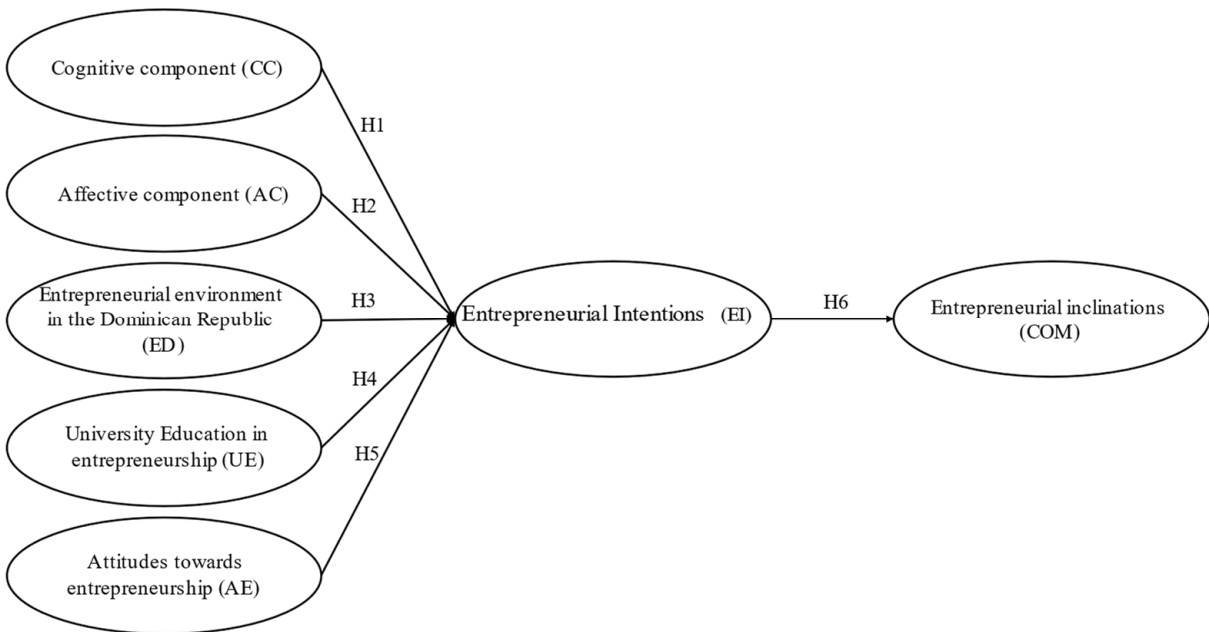

**Figure 1.** Proposed structural model. Source: Prepared by the authors.

### 3. Methodology

#### 3.1. Context of the Study

This research focused on the Dominican Republic. In the last decade, the country has had one of the fastest growing economies in Latin America and the Caribbean [48], which has led to the creation of new businesses [2,3]. In turn, many universities in the country have promoted entrepreneurship training as a transversal educational offer, including this training in all study programmes. The Universidad Tecnológica de Santiago (UTESA) has been the institution selected to carry out this study for the following reasons: (1) it is the largest private university in the Dominican Republic (and second largest overall) in terms of number of graduates (+138,000), active students (+40,000) and administrative and academic employees (+2000); (2) it is a face-to-face university, but is located in seven provinces of the country (Santo Domingo, Santiago de los Caballeros, Moca, Mao, Dajabón, Puerto Plata and Gaspar Hernández) (Figure 2); (3) it has a broad undergraduate offering, with more than 30 programmes, and offers Master's and Doctoral programmes; (4) it has an entrepreneurial spirit, being the only university in the country that has a corporate system where, in addition to the university, there are companies (a medical centre, several medical clinics, a hotel, a newspaper, a free trade zone, agricultural companies, a convention and culture centre and a radio-television programme), where students carry out their university internships in relation to the degree they are studying; (5) and, finally, because all study plans offer three compulsory subjects for all students, to be taught in the last year of the degree course: "Entrepreneurship Training," "Thesis Proposal" and "Thesis." In "Entrepreneurship Training," students learn about different tools for analysing business ideas; in the "Thesis Proposal" subject, students carry out market studies to find out about different aspects of the ideas identified in the "Entrepreneurship Training" subject; lastly, in the "Thesis" subject, students carry out technical and financial analyses to find out about the viability of the business idea they have identified. This compulsory training has meant that many students have applied for funding for business projects in the country and have been successful, which has meant that today they have a business in operation.

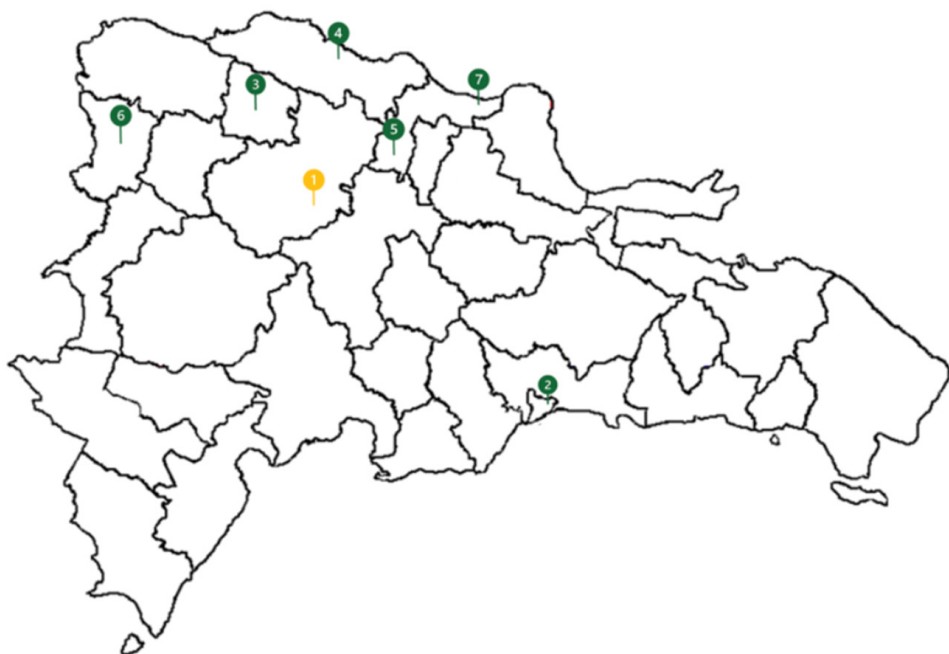

**Figure 2.** Areas where the UTESA university is located. Source: Prepared by the authors.

*3.2. Measurements*

The theoretical constructs of this research were assessed using a five-point Likert-type scale (1 = strongly disagree, 3 = neither disagree nor agree; 5 = strongly agree). The appropriate scales were adapted based on a review of the relevant literature [1,49,50]. A five-step procedure was followed to adapt the original scales to Spanish. First, two native Spanish-speaking translators (Dominicans) carried out the direct translation from English into Spanish. The two translations were then compared, and a preliminary draft was produced. The preliminary draft was translated from Spanish into English by a native English-speaking translator. All translations made during the process were checked, and the final version of the survey was designed in Spanish. To ensure the comprehension of the questionnaire and the appropriateness of its structure, a pilot test was carried out with 28 students taking the subject "Thesis," and no problems were detected. Simple and concise language was used, avoiding syntactic complexity to mitigate possible biases [51]. In addition, respondents' anonymity was guaranteed; it was explained that there were no right or wrong answers, and the questionnaire was kept as short as possible to encourage accurate responses [51].

*3.3. Data Collection and Sample Profile*

The data collection was carried out by means of a structured self-administered questionnaire in Spanish, which was physically distributed to the students who were taking the subject "Thesis." Therefore, all the students had basic knowledge of entrepreneurship, as they had taken and passed the subjects "Entrepreneurship Training" and "Thesis Proposal." The population consisted of 7311 students enrolled in the subject "Thesis." From September 2022 to January 2023 inclusive, trained interviewers distributed and, where necessary, assisted respondents in completing the questionnaire. A sample of 523 questionnaires was obtained, which establishes a sampling error of ±4.13%. The sample consisted of female students (71.3%), aged between 22 and 24 (41.4%), working (67.1%) and earning less than US$500 per month (83.8%), with a household size between 2 and 4 persons (70.5%).

*3.4. Verification Strategy and Preliminary Data Analysis*

The researchers tabulated the data and carried out quality checks to ensure the validity of the data before testing the hypotheses. First, outliers and incorrect responses (e.g.,

answering the same item with several options) were identified, resulting in the elimination of 11 questionnaires, leaving a total of 523 valid questionnaires, as mentioned above. Secondly, preliminary item analysis (Table 1) was carried out using SPSS software (v.28.0).

**Table 1.** Variables used in the model.

| | | **Mean** | **SD** | **Norm.** | **Cronbach** |
|---|---|---|---|---|---|
| | Entrepreneurial inclinations (BE) | | | | 0.875 |
| BE1 | I enjoy the lectures on entrepreneurship provided at the university | 3.83 | 1.210 | 0.000 [C] | |
| BE2 | The lectures on entrepreneurship I received at the university have increased my interest in pursuing an entrepreneurial career | 3.87 | 1.173 | 0.000 [C] | |
| BE3 | I consider entrepreneurship as a very important subject at the university | 4.50 | 0.889 | 0.000 [C] | |
| BE4 | The entrepreneurial subjects I have taken at university have prepared me to make decisions to pursue an entrepreneurial career. | 3.88 | 1.132 | 0.000 [C] | |
| BE5 | I am happy to have had a business education at my university | 4.13 | 1.133 | 0.000 [C] | |
| BE6 | I sincerely consider entrepreneurship as a desired career option | 4.23 | 0.935 | 0.000 [C] | |
| BE7 | The entrepreneurship education I have received at university will encourage me to venture into entrepreneurship after graduation. | 4.02 | 1.066 | 0.000 [C] | |
| BE8 | My entrepreneurship teachers have helped me to meet and interact with successful entrepreneurs. | 3.71 | 1.208 | 0.000 [C] | |
| BE9 | The entrepreneurship staff at my university helps students meet successful entrepreneurs who motivate them to become entrepreneurs. | 3.59 | 1.221 | 0.000 [C] | |
| | Cognitive component (CC) | | | | 0.937 |
| CC1 | The entrepreneurship courses have enabled me to identify business-related opportunities. | 4.01 | 1.042 | 0.000 [C] | |
| CC2 | Entrepreneurship subjects have taught me how to create services and/or products that can meet the needs of customers. | 3.88 | 1.138 | 0.000 [C] | |
| CC3 | The entrepreneurship courses have taught me how to develop successful business plans. | 3.82 | 1.137 | 0.000 [C] | |
| CC4 | Due to entrepreneurship subjects, I now have skills to create a new business. | 3.98 | 1.142 | 0.000 [C] | |
| CC5 | With entrepreneurship subjects, I can now successfully identify sources of business opportunities. | 3.95 | 1.037 | 0.000 [C] | |
| CC6 | The entrepreneurship courses have taught me how to carry out feasibility studies. | 3.85 | 1.088 | 0.000 [C] | |
| CC7 | Entrepreneurship subjects have stimulated my interest in entrepreneurship. | 4.08 | 1.087 | 0.000 [C] | |
| CC8 | Through entrepreneurship subjects, my skills, knowledge and interest in entrepreneurship have improved. | 4.11 | 1.047 | 0.000 [C] | |
| CC9 | Overall, I am very satisfied with the way entrepreneurship subjects are taught at my university. | 3.92 | 1.196 | 0.000 [C] | |

**Table 1.** *Cont.*

| | | Mean | SD | Norm. | Cronbach |
|---|---|---|---|---|---|
| **Affective component (AC)** | | | | | 0.762 |
| AC1 | I would like to be an entrepreneur after my studies. | 4.56 | 0.861 | 0.000 [C] | |
| AC2 | I am attracted by the idea of becoming an entrepreneur and working for myself. | 4.60 | 0.797 | 0.000 [C] | |
| AC3 | I really consider self-employment as something very important | 4.63 | 0.717 | 0.000 [C] | |
| AC4 | The entrepreneurship subjects at university have effectively prepared me to establish a career in entrepreneurship. | 3.94 | 1.124 | 0.000 [C] | |
| **Entrepreneurial intention (EI)** | | | | | 0.867 |
| EI1 | A career as an entrepreneur is attractive to me | 4.31 | 0.952 | 0.000 [C] | |
| EI2 | If I had the resources, I would like to start a business | 4.66 | 0.728 | 0.000 [C] | |
| EI3 | The people I care about would approve of my intention to become an entrepreneur. | 4.57 | 0.698 | 0.000 [C] | |
| EI4 | Most people who are important to me would approve of me becoming an entrepreneur. | 4.57 | 0.689 | 0.000 [C] | |
| EI5 | Being an entrepreneur gives me satisfaction | 4.49 | 0.824 | 0.000 [C] | |
| EI6 | Being an entrepreneur gives me more advantages than disadvantages | 4.38 | 0.848 | 0.000 [C] | |
| EI7 | I prefer to be an entrepreneur among several options | 4.33 | 0.902 | 0.000 [C] | |
| **Entrepreneurial environment in the Dominican Republic (ED)** | | | | | 0.775 |
| ED1 | The Dominican Republic is an excellent country to start a business | 3.67 | 1.114 | 0.000 [C] | |
| ED2 | Local government supports entrepreneurs | 3.11 | 1.198 | 0.000 [C] | |
| ED3 | It would be very difficult to raise the money to start a new business in the Dominican Republic. | 3.58 | 1.098 | 0.000 [C] | |
| ED4 | I know how to access the assistance I need to start a new business. | 3.48 | 1.153 | 0.000 [C] | |
| ED5 | I am aware of the programmes offered by the country to help people start businesses | 2.99 | 1.328 | 0.000 [C] | |
| **University education in entrepreneurship (EU)** | | | | | 0.862 |
| EU1 | The subject of business organization gave me new knowledge about entrepreneurship | 3.98 | 1.129 | 0.000 [C] | |
| EU2 | The entrepreneurship training course provided me with new knowledge about entrepreneurship. | 4.22 | 1.047 | 0.000 [C] | |
| EU3 | The undergraduate thesis proposal course gave me new knowledge about entrepreneurship. | 4.27 | 1.026 | 0.000 [C] | |
| EU4 | The graduate thesis proposal course gave me new knowledge about entrepreneurship. | 4.36 | 1.010 | 0.000 [C] | |
| **Attitudes towards entrepreneurship (AE)** | | | | | 0.881 |
| AE1 | If I had the opportunity, I would like to start a company | 4.63 | 0.745 | 0.000 [C] | |
| AE2 | Being an entrepreneur would give me great satisfaction | 4.58 | 0.747 | 0.000 [C] | |
| AE3 | Becoming an entrepreneur appeals to me | 4.52 | 0.796 | 0.000 [C] | |

Source: Prepared by the authors. [C]: Lilliephors significance correction

Preliminary analysis shows that the variables analysed do not follow normality assumptions, so a non-parametric test will be applied. The reliability analysis of the scale was carried out using Cronbach's alpha, obtaining overall values of 0.957 and 0.937 for the cognitive component; 0.762 for the affective component; 0.775 for the entrepreneurial environment in the Dominican Republic; 0.862 for university training in entrepreneurship; 0.881 for attitudes towards entrepreneurship; 0.867 for entrepreneurial intention; and 0.875 for entrepreneurial behaviour. The figures obtained are well above the minimum required by reference authors [52], so the reliability of the scale is optimal.

Once the reliability of the items was tested, and in order to evaluate the hypotheses through structural equation modelling, PLS-SEM, a composite-based approach, was used, which focuses on predicting hypothesised relationships that maximise the variance explained in the dependent variables [53]. PLS-SEM is particularly appropriate when research focuses on predicting and explaining variance of key constructs [54], as it shows almost no bias [55]. Through PLS-SEM, greater predictive power is obtained with $R^2$ values, and more accurate effect sizes are presented [56]. Initially, the measurement model was conducted to test the reliability and validity of the constructs, and then the structural model was run to test the hypotheses [57]. In this regard, the SmartPLS software (v.3.3.7) was used [58].

## 4. Results

### 4.1. Analysis of the Measurement Model

The analysis of the measurement model is shown in Table 2. The reliability of individual items was assessed using factor loadings, where values above 0.707 imply that the shared variance between the construct and its indicators is greater than the error variance [59]. The internal consistency of the construct was tested using the composite reliability measure [60] because it is less frequently affected by common method bias [61]. Composite reliability for both the Dijkstra-Henseler coefficient (r_A) and the Dillon-Goldstein coefficient (r_C) have optimal values of 0.80 and above [57]. All constructs in this study exceed this value, which demonstrates their reliability. To assess convergent validity, the average variance extracted (AVE) value was calculated for each construct, all values being above the threshold of 0.50 [62].

**Table 2.** Validity and reliability analysis of the measurement model at the indicator level.

| Indicators/Compounds | Loads (Sig.) | r_A | rC | AVE |
|---|---|---|---|---|
| Affective component (AC) | | | | |
| AC1 | 0.873 | | | |
| AC2 | 0.901 | 0.844 | 0.871 | 0.638 |
| AC3 | 0.845 | | | |
| AC4 | 0.511 | | | |
| Cognitive component (CC) | | | | |
| CC1 | 0.795 | | | |
| CC2 | 0.799 | | | |
| CC3 | 0.813 | | | |
| CC4 | 0.853 | | | |
| CC5 | 0.824 | 0.942 | 0.948 | 0.667 |
| CC6 | 0.799 | | | |
| CC7 | 0.828 | | | |
| CC8 | 0.847 | | | |
| CC9 | 0.792 | | | |
| Entrepreneurial environment in the Dominican Republic (ED) | | | | |
| ED1 | 0.786 | | | |
| ED2 | 0.811 | 0.816 | 0.842 | 0.521 |
| ED3 | 0.528 | | | |

**Table 2.** *Cont.*

| Indicators/Compounds | Loads (Sig.) | r_A | rC | AVE |
|---|---|---|---|---|
| ED4 | 0.750 | | | |
| ED5 | 0.699 | | | |
| Entrepreneurial intention (EI) | | | | |
| EI1 | 0.759 | | | |
| EI2 | 0.742 | | | |
| EI3 | 0.659 | | | |
| EI4 | 0.696 | 0.880 | 0.899 | 0.561 |
| EI5 | 0.867 | | | |
| EI6 | 0.761 | | | |
| EI7 | 0.740 | | | |
| Attitudes towards entrepreneurship (AE) | | | | |
| AE1 | 0.873 | | | |
| AE2 | 0.900 | 0.884 | 0.926 | 0.808 |
| AE3 | 0.923 | | | |
| Entrepreneurial inclinations (BE) | | | | |
| BE1 | 0.720 | | | |
| BE2 | 0.762 | | | |
| BE3 | 0.630 | | | |
| BE4 | 0.753 | | | |
| BE5 | 0.818 | 0.877 | 0.898 | 0.515 |
| BE6 | 0.614 | | | |
| BE7 | 0.780 | | | |
| BE8 | 0.627 | | | |
| BE9 | 0.769 | | | |
| University education in entrepreneurship (EU) | | | | |
| EU1 | 0.841 | | | |
| EU2 | 0.868 | 0.863 | 0.906 | 0.708 |
| EU3 | 0.849 | | | |
| EU4 | 0.806 | | | |

Source: Author's own work.

Table 3 shows the discriminant validity analysis, carried out through the heterotrait-monotrait ratio (HTMT), which must obtain values lower than 0.90 [63]. In this research, all values are lower than recommended. HTMT analysis was used mainly because leading authors in the field [64] indicated that lack of discriminant validity was better detected through the heterotrait-monotrait ratio compared to the Fornell-Larcker criterion or cross-loadings.

**Table 3.** Discriminant validity. Heterotrait-monotrait ratio.

| | AC | AE | CC | BE | ED | EI | EU |
|---|---|---|---|---|---|---|---|
| | | | **Discriminant Validity (HT-MT Ratio)** | | | | |
| AC | | | | | | | |
| AE | 0.743 | | | | | | |
| CC | 0.640 | 0.388 | | | | | |
| BE | 0.695 | 0.433 | 0.885 | | | | |
| ED | 0.388 | 0.258 | 0.619 | 0.560 | | | |
| EI | 0.872 | 0.857 | 0.493 | 0.562 | 0.379 | | |
| EU | 0.627 | 0.429 | 0.848 | 0.797 | 0.555 | 0.530 | |

Key: AC: Affective component; CC: Cognitive component; ED: Entrepreneurial environment in the Dominican Republic; EI: Entrepreneurial intention; AE: Attitudes towards entrepreneurship; BE: Entrepreneurial inclinations; EU: University education in entrepreneurship. Source: Prepared by the authors.

Once the reliability and validity of the measurement model had been tested at both the individual and composite level, the structural model was analysed.

### 4.2. Analysis of the Structural Model

Before testing the hypotheses, Table 4 shows the results of the model in terms of predictive power, predictive relevance and explained variance. The results obtained show a significant and moderate predictive power of the endogenous variables that make up the model [57]. Thus, the effect of the antecedent variables on the endogenous variable (entrepreneurial intention -EI-) differs between a large and significant effect of the variable attitudes towards entrepreneurship (AE) and a moderate effect of the affective component variable (AC). The rest of the antecedent variables presented non-significant effects on the endogenous variable entrepreneurial intentions (EI). Moreover, a large and significant effect of entrepreneurial intention (EI) on its endogenous variable entrepreneurial inclinations (BE) can be observed [65]. As a result, the affective component variable is responsible for 29.35% of the variability of the endogenous variable entrepreneurial intention, and the variable attitude towards entrepreneurship is responsible for 35.38% of the variability of the endogenous variable entrepreneurial intention. It is also observed that the entrepreneurial intention variable explains 27.77% of the variance of the entrepreneurial inclinations variable.

**Table 4.** Predictive power and explained variance.

| Hypothesis | b | $R^2$ | $f^2$ (Sig.) | Correlation | Explained Variance |
|---|---|---|---|---|---|
| EI | | 0.710 | | | |
| H1: CC | 0.002 | | 0.000 (0.999) | 0.460 | −0.092% |
| H2: AC | 0.394 | | 0.276 (0.006) | 0.745 | 29.35% |
| H3: ED | 0.080 | | 0.016 (0.289) | 0.340 | 2.72% |
| H4: EU | 0.077 | | 0.008 (0.498) | 0.471 | 3.62% |
| H5: AE | 0.465 | | 0.441 (0.000) | 0.761 | 35.38% |
| BE | | | | | |
| H6: EI | 0.527 | 0.277 | 0.384 (0.000) | 0.527 | 27.77% |

Source: Prepared by the authors.

To test the structural model, the bootstrapping technique has been used [57], obtaining pcoefficients associated with a limit probability and a *t* statistic. Similarly, the results are offered from a non-parametric perspective (via hypothesis contrast) in Table 5. The results obtained show the influence of the affective component (H$_2$), the country's business environment (H$_3$) and attitude towards entrepreneurship (H$_5$) on entrepreneurship intention. Furthermore, the rest of the hypotheses raised (H$_1$ and H$_4$) have not been supported, and, therefore, the influence of the cognitive component and university training in entrepreneurship on entrepreneurial intention has not been confirmed. The final structural model is presented in Figure 3.

**Table 5.** Hypothesis testing.

| Hypothesis Put Forward | b | t (p.lim.) | IC95% | |
|---|---|---|---|---|
| | | | 2.5% | 97.5% |
| H1: CC → EI | −0.002 [NS] | 0.041 (0.967) | −0.087 | 0.086 |
| H2: AC → EI | 0.394 *** | 6.471 (0.000) | 0.273 | 0.511 |
| H3: ED → EI | 0.080 ** | 2.341 (0.019) | 0.017 | 0.152 |
| H4: EU → EI | 0.077 [NS] | 1.49 (0.135) | −0.023 | 0.179 |
| H5: AE → EI | 0.465 *** | 9.431 (0.000) | 0.369 | 0.561 |
| H6: EI → BE | 0.527 *** | 12.941 (0.000) | 0.449 | 0.607 |

Source: Prepared by the authors. **: *p* < 0.01; ***: *p* < 0.001; [NS]: Not Supported.

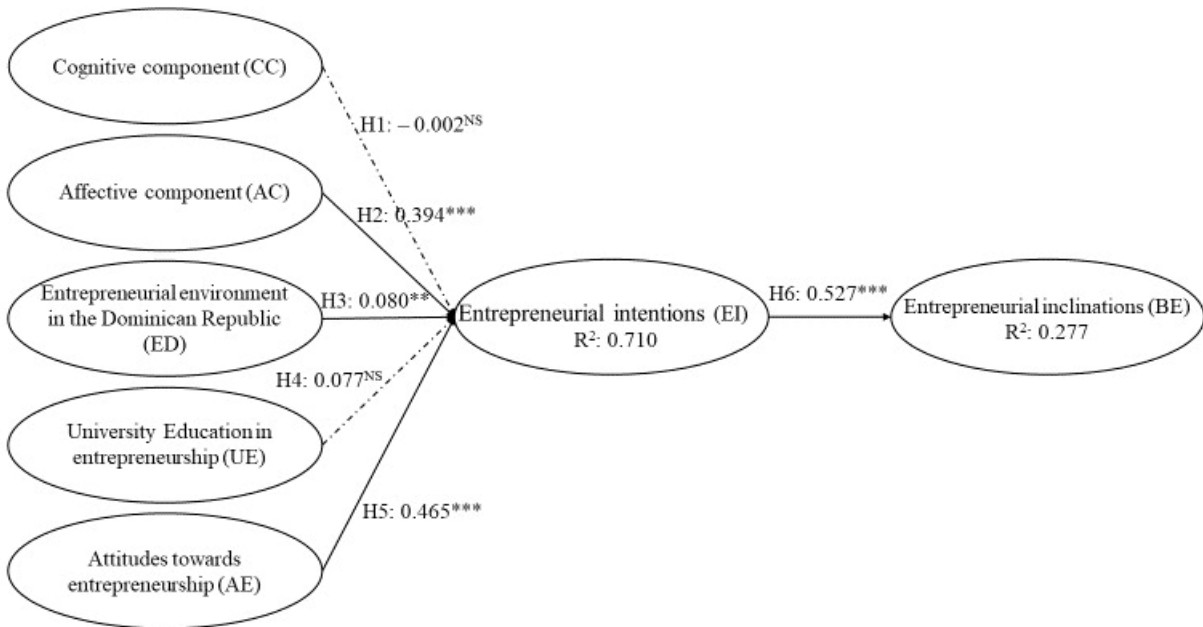

**Figure 3.** Final structural model. Source: Prepared by the authors. **: $p < 0.01$; ***: $p < 0.001$; NS: Not Supported.

## 5. Discussion and Conclusions

The results of this research indicate that different factors have varying degrees of influence on the entrepreneurial intention of university students. The affective component, which refers to the emotional aspect of attitudes towards entrepreneurship, explains a significant part (29.35%) of the variance in entrepreneurial intention. This suggests that students' emotional connection and positive feelings towards entrepreneurship play a crucial role in shaping their intention to become entrepreneurs [66]. Attitudes towards entrepreneurship, on the other hand, explain an even greater portion (35.38%) of the variance in entrepreneurial intention. This is in line with other studies [1,67,68], where it had already been indicated that the influence of attitudes on entrepreneurial intention has a high explanatory power and is extremely relevant for increasing students' entrepreneurial intentions. This highlights the importance of students' attitudes towards entrepreneurship in their entrepreneurial intentions. Thus, positive attitudes towards entrepreneurship can serve as a driving force for students to actively consider and pursue entrepreneurial opportunities.

Furthermore, the country's business environment, which encompasses factors such as economic conditions, government policies and market opportunities, explains a relatively smaller portion (2.72%) of the variation in entrepreneurial intention. This suggests that while the external environment may have some influence, it is not the main determinant of students' entrepreneurial intention. Jena [1] did find a relationship between a country's business environment and students' entrepreneurial intentions; however, the relationship may be conditioned by country-specific factors. Furthermore, the results indicate that entrepreneurial intention itself has a substantial influence (27.77%) on entrepreneurial inclinations. This suggests that students who have a strong intention to become entrepreneurs are more likely to engage in entrepreneurial activities and inclinations, which is consistent with other studies [1,21]. This finding aligns with the theory that intention is a crucial precursor to actual behaviour.

The hypotheses related to the cognitive component and university training on entrepreneurship were not supported in this study. This suggests that factors such as cognitive beliefs and knowledge acquired through university education may not have a direct impact on students' entrepreneurial intention; however, other studies have found a relationship between these variables [1,37,38]. These results therefore highlight the complex nature of entrepreneurial intention in students and the multifaceted factors that contribute to it.

### 5.1. Suggestions

The findings of this research highlight the importance of the affective component and attitudes towards entrepreneurship in the formation of students' entrepreneurial intentions. Emphasising the emotional aspect of attitudes and promoting positive feelings towards entrepreneurship can play a crucial role in fostering students' intention to become entrepreneurs. Moreover, the strong influence of attitudes towards entrepreneurship on entrepreneurial intentions reinforces the importance of cultivating positive attitudes through educational interventions. These results contribute to the understanding of the drivers of entrepreneurial intentions among university students and provide a basis for future research in this area.

The practical implications of this research suggest several strategies for promoting entrepreneurial intentions among university students. Universities should design programmes that focus on the affective component, focusing on fostering positive emotions and developing an entrepreneurial mindset. In addition, the integration of practical experiences, such as internships and mentoring programmes, can provide students with real-world exposure to entrepreneurship and enhance their intentions to engage in entrepreneurial activities. Education and enterprise policy makers should consider developing supportive policies and creating a favourable business environment to foster entrepreneurship. However, it is important to recognise that the influence of the external environment may be relatively limited compared to factors at the individual level. These findings provide actionable information for stakeholders involved in fostering entrepreneurial intentions among university students, guiding the design of effective entrepreneurship education programmes and policies.

In this context, the development of public policies in favour of entrepreneurship ecosystems could increase the intention of entrepreneurship [69], and students may benefit if Dominican universities are included as a fundamental part, above all, in those rural areas where there are more difficulties in training in entrepreneurship [70]. In this regard, both the Ministry of Higher Education, Science and Technology (MESCyT) and the Ministry of Industry, Commerce and MSMEs would play a fundamental role.

### 5.2. Limitations and Future Research Lines

The main drawback of this research is its cross-sectional nature, so we cannot attribute causality to the observed relationship. Therefore, conducting further longitudinal studies is essential to confirm the relationships observed in this study. The study is conducted within the university context of the Dominican Republic, but solely from the viewpoint of a university. Future research can be developed considering other universities and a wider sample. This would help in the generalization of the results. It is possible that other factors not considered in this study may influence, mediate or moderate the relationship between cognitive factors and entrepreneurial intention. Future research could explore additional factors and possible interactions between different variables to gain a more complete understanding of the determinants of entrepreneurial intention in students.

**Author Contributions:** Conceptualization, M.R.V.-M. and C.M.D.-V.; methodology, M.R.V.-M. and C.M.D.-V.; software, M.R.V.-M. and C.M.D.-V.; validation, C.M.D.-V. and N.d.J.C.-F.; formal analysis, C.M.D.-V.; investigation, M.R.V.-M.; resources, M.R.V.-M.; data curation, N.d.J.C.-F.; writing—original draft preparation, M.R.V.-M.; writing—review and editing, J.C.T.-D.H., N.d.J.C.-F. and C.M.D.-V.; visualization, J.C.T.-D.H. and N.d.J.C.-F.; supervision, J.C.T.-D.H.; project administration, M.R.V.-M. All authors have read and agreed to the published version of the manuscript.

**Funding:** This research received no external funding.

**Institutional Review Board Statement:** Not applicable.

**Informed Consent Statement:** Not applicable.

**Data Availability Statement:** Data are available upon request from researchers.

**Acknowledgments:** The researchers are grateful for the support received from both the Technological University of Santiago (Dominican Republic) and the researchers from the Department of Statistics, Econometrics, Operations Research, Business Organization and Applied Economics of the University of Córdoba (Spain).

**Conflicts of Interest:** The authors declare no conflict of interest.

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
