# Peer review of "Business Environment, Attitudes and Entrepreneurial Intentions as Antecedents of Entrepreneurial Inclination among University Students"

_sustainability, doi:10.3390/su151612280_

Round 1
Reviewer 1 Report
I like the Dominican Republic setting and the description of the provision of entrepreneurship education in the country – I believe this will interest readers. The paper is laid out clearly and the authors recognise some of the study’s limitations. I think for me, the main concern was the measures used (see more detailed comments below) which don’t in some places seem to reflect what is actually being measured (lacking face validity). I would advise authors to reflect and then comment on how the measures were derived ensuring at least their face validity. To the authors’ credit, they are transparent with regard to the individual items that make up the measures so the reader is at least able to pass judgement themselves on their suitability. I would also like to see a little more grounding in some of the key literature on entrepreneurial intentions and a clearer distinction between antecedents of intent, intent and entrepreneurial behaviour. These all seem to coalesce into one in this study.
Here some further pointers:
I would avoid claims that cognitive and affective components have not been studied before in entrepreneurial intent studies. There must be over 100 ent. intent studies so the authors are standing on a substantial body of knowledge (see for example Bae et al. or Nabi et al.’s studies:
Bae, T. J., Qian, S., Miao, C., & Fiet, J. (2014). The Relationship Between Entrepreneurship Education and Entrepreneurial Intentions: A Meta-Analytic Review. Entrepreneurship Theory and Practice, 38(2), 217-254. https://doi.org/10.1111/etap.12095
Nabi, G., Liñan, F., Fayolle, A., Krueger, N., & Walmsley, A. (2017). The impact of entrepreneurship education in higher education: A systematic review and research agenda. . Academy of Management Learning and Education, 16(2), 277-299. https://doi.org/doi:10.5465/amle.2015.0026
Absolute statements such as ‘this is the only study that does X,Y,Z…’ should not be made unless the authors are absolutely convinced they have covered all the literature in their field which I don’t believe has occurred here.
I’d also recommend the authors look at
Souitaris, V., Zerbinati, S., & Al-Laham, A. (2007). Do entrepreneurship programmes raise entrepreneurial intention of science and engineering students? The effect of learning, inspiration and resources. Journal of Business Venturing, 22(4), 566-591.
The dependent variable ‘Entrepreneurial Behaviour’ is largely made up of components that reflect attitudes and intentions rather than behaviour. To call this behaviour is therefore a little misleading. Perhaps rephrase to ‘entrepreneurial inclination’
I am also not sure that the affective components actually measure affect? How do they differ from some of the components of the dependent variable, e.g. AC1 = ‘I would like to be an entrepreneur when I grow up’ – isn’t this a measure of entrepreneurial intent? Similarly, with regard to entrepreneurial intention, some of its components would align with subjective norms or attitudes (see Ajzen’s Theory of Planned Behaviour which has been applied in many studies of entrepreneurial intent) which then lead to intent rather than being intent itself (e.g. EI3 “The People I care about would approve of my intention to become an entrepreneur“ – this is clearly a measure of social norms based on TPB).
I’d advise the authors to be more judicious in the application of references. For example, it is claimed that the Jena et al. (2020) reference (reference no.1) suggests business education is a key determinant of a country’s economic growth/competitiveness but the title of Jena et al.’s (2020) paper makes it clear their paper does not look at this at all. I don’t know of any studies that would have made this kind of claim (i.e. business education is a key determinant for a country’s economy/competitiveness) and so this is what drew me to the reference, where I then found the reference didn’t really align with what the authors are claiming. I am also not sure about the application of reference no.5 (Smith et al. 2020) on page 1 with reference to the definition of entrepreneurship (not all entrepreneurs are out to seek purely profit, and even less so continuous growth – consider for example lifestyle entrepreneurs).
This was on the whole well written - I do not see the need for any major changes here apart from a final proof read.
Author Response
Dear reviewer. Thank you very much for his comments. We have attached a document with the responses to your comments, and we have included a new version of the document with the suggestions of both reviewers in blue. We remain attentive to any other suggestion or question. Thank you.

Reviewer 2 Report
I congratulate the authors for this study, which I think will contribute to researchers. I think the suggested corrections will improve the article.

Author Response

(The authors gave the same response as above.)

Round 2
Reviewer 1 Report
Dear Authors, thank you for reflecting on and addressing my comments. I hope you found these helpful and have resulted in an improved manuscript. As mentioned in my original review, studies such as yours undertaken in non-typical contexts should add to our understanding of the development of entrepreneurial intentions (and inclination) at a global level.